# Co-translational insertion and topogenesis of bacterial membrane proteins monitored in real time

Evan Mercier, Wolfgang Wintermeyer & Marina V Rodnina[*]

## Abstract

Integral membrane proteins insert into the bacterial inner membrane co-translationally via the translocon. Transmembrane (TM) segments of nascent proteins adopt their native topological arrangement with the N-terminus of the first TM (TM1) oriented to the outside (type I) or the inside (type II) of the cell. Here, we study TM1 topogenesis during ongoing translation in a bacterial in vitro system, applying real-time FRET and protease protection assays. We find that TM1 of the type I protein LepB reaches the translocon immediately upon emerging from the ribosome. In contrast, the type II protein EmrD requires a longer nascent chain before TM1 reaches the translocon and adopts its topology by looping inside the ribosomal peptide exit tunnel. Looping presumably is mediated by interactions between positive charges at the N-terminus of TM1 and negative charges in the tunnel wall. Early TM1 inversion is abrogated by charge reversal at the N-terminus. Kinetic analysis also shows that co-translational membrane insertion of TM1 is intrinsically rapid and rate-limited by translation. Thus, the ribosome has an important role in membrane protein topogenesis.

**Keywords** *In vitro* translation; positive-inside rule; real-time translation kinetics; ribosome; transmembrane segment inversion

**Subject Categories** Membranes & Trafficking; Translation & Protein Quality

**The EMBO Journal (2020) 39: e104054**

## Introduction

The majority of integral membrane proteins are inserted into the membrane co-translationally. Ribosomes synthesizing membrane proteins are targeted to the protein-conducting channel (translocon) of the bacterial plasma membrane or the endoplasmic reticulum in eukaryotes by the signal recognition particle (SRP), which recognizes signal or signal-anchor sequences near the N-terminus of the nascent peptide (Cymer *et al*, 2015; Kuhn *et al*, 2017). The bacterial core translocon is a ternary complex consisting of proteins SecY, SecE, and SecG inserted into the plasma membrane (Bischoff *et al*,

2014; Park *et al*, 2014). The major translocon component, SecY, comprises 10 transmembrane (TM) segments that are arranged in a pseudosymmetrical structure with TM segments 1–5 and 6–10 forming a central pore through which secretory proteins can pass into the periplasm (Van den Berg *et al*, 2004; Tanaka *et al*, 2015). The two halves of the translocon can also move apart to open laterally (Egea & Stroud, 2010; Voorhees & Hegde, 2016). Lateral gate opening is induced by the binding of ribosome–nascent chain complexes (RNCs; Ge *et al*, 2014; Kater *et al*, 2019). Once the lateral gate is opened, TM segments of the nascent chain can partition between the hydrophilic inner pore of the translocon and the surrounding hydrophobic phospholipid bilayer.

Transmembrane segments can insert into the membrane in two orientations, with the N-terminus pointing either outwards into the periplasm of bacteria or the ER lumen in eukaryotes (N-out or type I topology) or inwards into the cytoplasm (N-in or type II topology). The orientation of the TM segment of single-spanning membrane proteins or TM1 of multi-spanning proteins generally is determined by the distribution of charged amino acids (aa) at the N-terminus of the TM; positively charged segments are retained on the cytoplasmic side of the membrane ("positive-inside rule"; von Heijne, 1989). The N-in orientation requires that the TM segment is inverted at some point during synthesis or membrane insertion. Where and when the inversion takes place, i.e., prior to, during, or following insertion into the translocon, are not clear. During translation, nascent proteins traverse the polypeptide exit tunnel of the ribosome, which spans about 100 Å from the peptidyl transferase center (PTC) to the peptide exit port where the translocon binds (Voss *et al*, 2006; Frauenfeld *et al*, 2011). Within the exit tunnel, proteins can fold into α-helices or even into small α-helical domains (for references, see ref. Rodnina, 2016). Depending on the nascent peptide fold, the exit tunnel can occlude a minimum of 30 aa (assuming a fully extended conformation) and up to 70 aa (assuming a folded α-helix or a partially folded protein). The transmembrane segments of the translocon channel, which span additional 35–40 Å, form a conduit with the ribosome exit tunnel and can occlude about 25 aa of the TM segment (Hildebrand *et al*, 2004; Frauenfeld *et al*, 2011). Studies on eukaryotic model proteins identified an N-out intermediate en route to an N-in membrane protein and suggest that the choice of the N-in or N-out topology is a late event (Goder &

Department of Physical Biochemistry, Max Planck Institute for Biophysical Chemistry, Göttingen, Germany
*Corresponding author. Tel: +49 551 201 2900; E-mail: rodnina@mpibpc.mpg.de

Spiess, 2003; Devaraneni *et al*, 2011). Alternatively, the interaction of positive charges at the N-terminus of the TM segment with negative charges of membrane phospholipids might cause retention and initiate TM inversion at or in the translocon. In fact, mutations in the Sec61 translocon in yeast have an effect on signal sequence recognition and the topology of membrane insertion, consistent with an important role of the translocon in determining the topology of membrane proteins (Goder *et al*, 2004). For the bacterial system, not much is known about molecular details of TM inversion, and data on co-translational membrane insertion are not available.

The major question we address in the present work is at which point retention and inversion of the N-in type nascent chain take place in a bacterial system. To be able to monitor co-translational membrane insertion in a physiologically relevant time frame, we have established an *in vitro* translation system reconstituted from purified components that performs at near *in vivo* rate and accuracy (Rudorf *et al*, 2014; Holtkamp *et al*, 2015). To study nascent chain membrane insertion via the translocon, we use the *E. coli* SecYEG core translocon embedded into *E. coli* membrane phospholipids contained within nanodiscs. Nanodiscs are planar phospholipid bilayer discs held together by a membrane scaffold protein (such as MSP1D1) derived from apolipoprotein A1 of high-density lipoproteins (Alami *et al*, 2007). This allows highly purified, biochemically well-defined translocons to be studied in a native-like membrane

environment. Nanodisc-embedded translocons are functional in protein translocation and form high-affinity complexes with ribosomes (Ge *et al*, 2014). The FRET efficiency at different nascent chain lengths is used as a ruler for the movement of the nascent chain relative to the translocon, which provides insights into translocon insertion and protein topogenesis. By using both FRET and proteolysis approaches, we observe that the N-in type of insertion requires a longer nascent chain than the N-out type. This reflects looping of the N-in nascent chain accompanying topological inversion within the peptide exit tunnel of the ribosome. Furthermore, the comparison of time courses of translation and translocon insertion reveals that insertion is rate-limited by translation, which implies that insertion is intrinsically rapid.

## Results

### Monitoring co-translational TM insertion by FRET

To follow the synthesis and topogenesis of nascent proteins in real time, we placed FRET reporters at the N-terminus of the emerging nascent chains of inner-membrane proteins and at either cytoplasmic or periplasmic loops of SecY (Fig 1A). When the growing nascent chain progresses from the PTC to the exit port of the

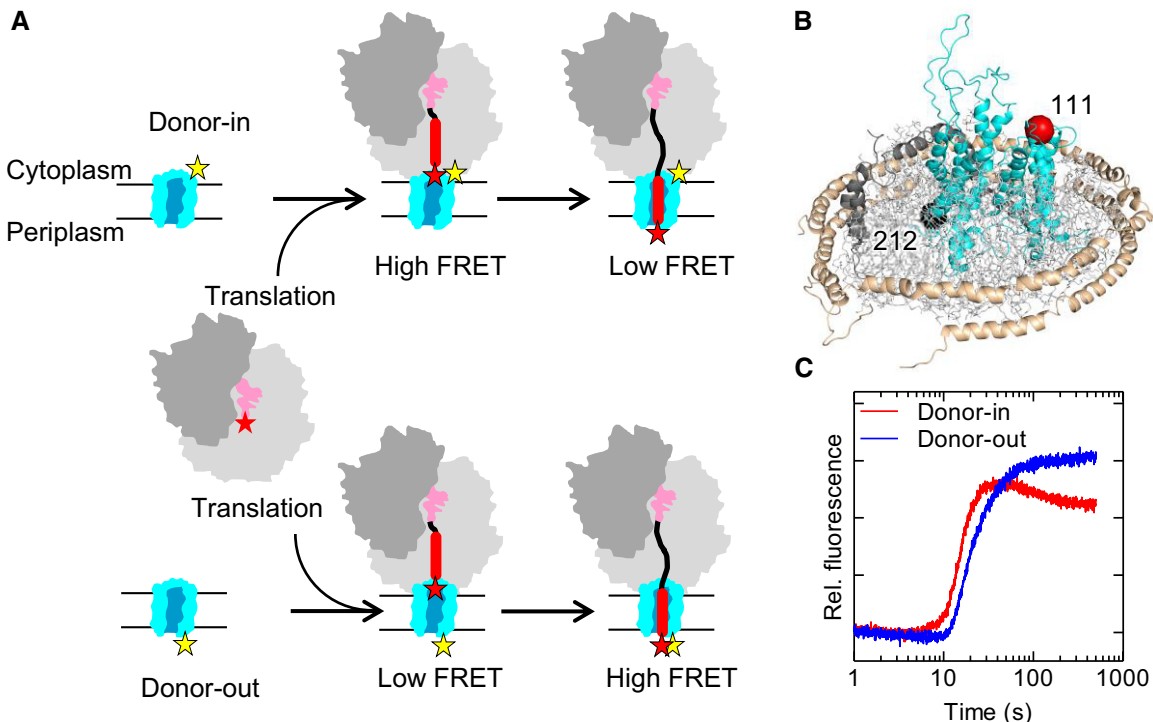

**Figure 1. Monitoring co-translational membrane protein insertion in real time by FRET.**

A  Expected FRET changes during co-translational insertion of an N-out membrane protein. The FRET acceptor (Atto655, red star) was placed at the N-terminus of the nascent chain, the FRET donor (Atto488, yellow star) at one of two positions in SecY (see panel B).

B  Positions on SecY (cyan) used for site-specific Atto488 donor labeling at the cytoplasmic face (position 111; red sphere) or periplasmic face (position 212; black sphere). (SecE, orange; SecG, green). The SecYEG translocon is embedded into phospholipids contained in a nanodisc held together by membrane scaffold proteins. Model based on PDB ID: 3J45.

C  Time-dependent acceptor fluorescence changes due to FRET during co-translational insertion of acceptor-labeled LepB75 into donor-labeled translocon.

ribosome–translocon complex, FRET efficiency is expected to increase as the N-terminus of the nascent peptide moves toward the translocon (Appendix Fig S1). For an N-out topology, such as that depicted in Fig 1A, we expect a co-translational increase in FRET when the N-terminus emerges from the exit tunnel in the proximity of the FRET reporter at the cytoplasmic side of the translocon. Subsequently, the N-terminus can move through the translocon away from the cytoplasmic side toward the periplasmic side, which should result in a lower FRET efficiency. Thus, in a time-resolved FRET experiment with the donor attached to the cytoplasmic face of SecY we would expect an increase from no-FRET to high-FRET efficiency, with a subsequent transition to a somewhat lower FRET efficiency. On the other hand, if the donor dye is attached to the periplasmic face of SecY, we expect to see only an increase in FRET as the nascent protein traverses the lengths of the exit tunnel and the translocon channel (Fig 1A).

To validate the FRET approach, we first examine the kinetics of co-translational translocon insertion of an N-out TM that lacks positively charged aa at the N-terminus and inserts into the membrane without inversion (Materials and Methods). As a model protein, we use leader peptidase (LepB) which has an N-terminal hydrophobic signal-anchor sequence (TM1) that inserts into the membrane in an N-out orientation (Facey & Kuhn, 2004) (Appendix Fig S2A). TM1 is followed by TM2 and the catalytic domain that resides in the periplasm (Paetzel *et al*, 1998). To study topogenesis of LepB TM1, we use a LepB75 mRNA construct coding for the two TM segments (Appendix Fig S2A) that is long enough to emerge from the peptide exit tunnel and to insert into the translocon when it is fully synthesized (Bornemann *et al*, 2008).

We placed a FRET donor (Atto488) on SecY either at the cytosolic face (position 111) or at the periplasmic face (position 212; Fig 1B; Materials and Methods). The nascent chain carries a FRET acceptor (Atto655) attached to the N-terminal methionine by using Atto655-Met-tRNA$^{fMet}$ for translation initiation. Translation starts upon rapid mixing of initiation complex with the components required for translation elongation (elongation factors EF-Tu, EF-G, and EF-Ts, aminoacyl-tRNA, and GTP) and nanodisc-embedded translocons. In this experimental setup, translation is synchronized because all ribosomes start elongation at the same time and there is no mRNA turnover due to the lack of termination factors and stop codon. Because the concentration of translocons is well above the $K_d$ value of 10–20 nM for ribosome–translocon complexes (Ge *et al*, 2014; Draycheva *et al*, 2016), and translocon binding to ribosomes is rapid, about 100 $\mu M^{-1} s^{-1}$ (A. Draycheva and W. Wintermeyer, unpublished data), all ribosomes translate while bound to translocons. Translocon binding does not affect the rate of translation (Appendix Fig S2B and C). We follow changes in FRET efficiency in real time in a stopped-flow apparatus by exciting the donor and monitoring the fluorescence of the acceptor.

When the FRET donor is placed on the cytosolic face of SecYEG ("donor-in"; Fig 1B), we observe a rapid FRET increase, starting after about 10 s of translation, which is followed by a slower FRET decrease (Fig 1C). This reflects the approach of the nascent chain N-terminus to the cytosolic side of the translocon, and the movement away toward the periplasmic side. Control time courses monitored by the fluorescence change of the FRET donor show the inverse change, as expected for FRET, albeit with a smaller relative fluorescence change due to the presence of an excess of donor-

labeled translocons (Appendix Fig S2D–F). When the FRET donor is attached to the periplasmic side of the translocon ("donor-out"), we observe a FRET increase following the delay, and no slow FRET decrease. As expected, the FRET increases earlier when the donor is placed at the cytoplasmic side compared to the periplasmic side (Fig 1C). These FRET changes are consistent with a model where N-out topology results from nascent chain insertion into the translocon in a head-first direction. This orientation is retained upon movement of the nascent chain toward the periplasmic side.

## Co-translational insertion of a type I TM (LepB)

To study the link between translation and nascent protein topogenesis, we first analyzed the kinetics of translation. We performed translation as described above, but stopped the reactions at the indicated incubation times, separated peptide products by SDS–PAGE, and visualized Atto655-containing products by fluorescence imaging (Fig 2A; Materials and Methods). Synthesis of LepB is generally rapid, and on SDS–PAGE, the final product migrates as a predominant band of the expected chain length. To test whether the translation rate varies along the mRNA, we performed experiments with mRNAs of different lengths, from LepB35 to LepB94 (Fig 2B, Appendix Fig S3). We estimated the average translation rate from the duration of the delay and an exponential term describing the time course of translation (Materials and Methods). The average translation rate ($k_{av}$) is about 1.5 aa/s, and there are small variations of the rate along the mRNA (Appendix Table S1). We also observe the transient accumulation of shorter peptides (P1 and P2), which is the hallmark of ribosome pausing (Mercier & Rodnina, 2018). Potential reasons for translation pauses are diverse and include a limiting amount of aminoacyl-tRNAs, the amino acid composition of the peptide, or the mRNA structure (Bevilacqua *et al*, 2016; Komar, 2016; Rodnina, 2016; Schuller & Green, 2018). Ribosome profiling shows several hot-spots of high ribosome density along the LepB mRNA, indicating that translational pausing takes place *in vivo* (Mohammad *et al*, 2019). *In vitro*, we observe pausing resulting in the accumulation of P1 already with the shortest mRNA construct tested, LepB35 (Appendix Fig S3). This agrees with an increase in ribosome density at codon 14 observed in profiling experiments (Mohammad *et al*, 2016). The P2 peptide, which is clearly distinguished on the translation gels of LepB94 (Fig 2B), is not found until after 65 aa chain length and appears very close to the nascent chain length of LepB75 (Appendix Fig S3). In profiling experiments, there is an accumulation of ribosomes at codon 68 (Mohammad *et al*, 2019). Thus, we assign the positions of transient ribosome pausing to codons 14 (P1) and 68 (P2).

To refine the kinetic analysis of LepB synthesis, we use a global fitting approach that we previously established for the analysis of translation and co-translational protein folding (Mercier & Rodnina, 2018). We use a kinetic model that starts with initiation complex and proceeds by elongating the nascent chain one aa at a time, with an elongation rate, $k_{el}$, assumed to be uniform for every aa (Fig 2C; Materials and Methods). Ribosome pausing is modeled as two off-pathway states at codons 14 and 68 that are reversibly connected to the elongation pathway and represent the minimum number of pauses needed for a satisfactory fit. The exact position of the early

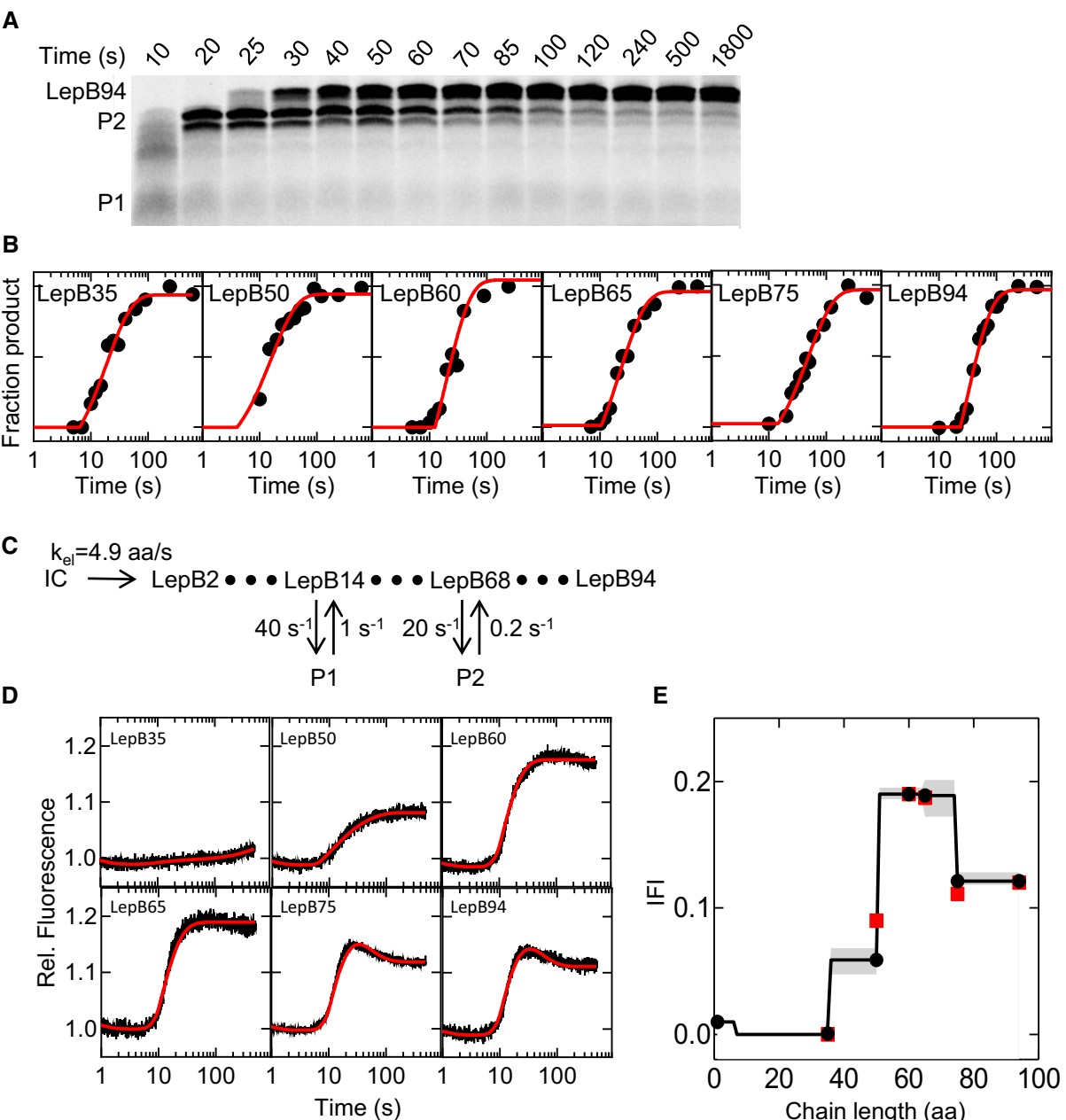

**Figure 2. Translation and topogenesis of LepB.**

A   Analysis of LepB94 translation products on SDS–PAGE. Translation products are visualized by the fluorescence of N-terminal Atto655. Pausing intermediates are indicated P1 and P2.

B   Fitting of time courses (filled circles) of the translation of LepB mRNA constructs of increasing length (Appendix Fig S3) by delay-exponential functions (red lines).

C   Kinetic model used for fitting of both translation and FRET time courses. Ribosome pausing at intermediates P1 and P2 is modeled as off-pathway states.

D   Stopped-flow time courses of co-translational LepB insertion into SecYEG (Atto488 donor-in at position 111 of SecY), as monitored by the fluorescence of the Atto655 acceptor placed at the N-terminus of the nascent peptide. Fits (red lines) were obtained by global fitting (Materials and Methods).

E   Intrinsic fluorescence intensities (IFI) derived from global fitting are shown as black lines and filled circles along with amplitudes obtained from exponential fitting (red squares). Confidence contours (shaded areas) were calculated using the FitSpace algorithm (Materials and Methods).

Source data are available online for this figure.

pause site at aa 14 is not important, as introducing a pause at any position below 30 aa results in the same fit and does not affect the evaluation of the FRET trajectories (see below). As for the second pause site, fitting of the translation kinetics necessitated an intermediate in the range between 65 and 75 aa; in fact, the FRET data are fitted best with a pause at aa 68, consistent with profiling data (Mohammad *et al*, 2016). We performed a global fit of all time courses together, using numerical integration, and determined the

elongation rate, $k_{el}$ = 4.9 aa/s, as well as the rates of reversible pausing (Fig 2C); the lower average rate of 1.5 aa/s is due to local pausing events. A more complex model that takes into account potential codon-specific rate differences yields a fit of comparable quality, as well as similar fits and pausing rates (Materials and Methods). Introducing varying codon-specific translation rates instead of reversible excursions to paused states does not yield satisfactory fits (not shown).

We next monitor FRET changes upon membrane insertion of nascent LepB using a FRET donor at the cytosolic face of the translocon (Fig 1A), as this label provides information on the movement of the nascent chain toward and through the translocon (Fig 1C). We use LepB mRNA constructs of increasing length, from LepB35 to LepB94. For the shortest construct, LepB35, the FRET signal is very small (Fig 2D), consistent with the expected large separation of donor (translocon) and acceptor (nascent chain). For LepB50, 60 and 65, the FRET increases, indicating a movement of the nascent chain toward the label at the cytosolic face of the translocon. The final levels for the three constructs indicate that the N-terminus is closer to the label on the translocon for LepB60 and 65, compared to LepB50. For the two longer constructs, LepB75 and 94, a late fluorescence decrease is evident (Fig 2D), indicating that the N-terminus of the nascent chain moves away from the cytosolic side of the translocon, presumably through the inner pore of the translocon. The FRET changes observed with LepB75 and LepB94 are very similar, suggesting that the nascent chain reaches a fixed orientation relative to the translocon at a length of 75 aa.

To determine whether the observed FRET changes are co- or post-translational, we fitted each time course with a single- or double-exponential function following a delay (Appendix Table S1). Comparison with the translation time courses analyzed on SDS–PAGE indicates that practically all fluorescence changes during LepB insertion occur co-translationally, including the fluorescence decrease observed for LepB75 and 94, with very small post-translational rearrangements observed for LepB35 only (Appendix Table S1). To reconstruct the trajectory of insertion, we first estimate relative FRET values from the endpoints of reactions for every LepB construct by exponential fitting (Fig 2E, red symbols), which illustrates a crude trajectory of nascent chain movement. We note that the pausing site at position 68 results in the accumulation of a high-FRET intermediate; this pause allows us to monitor the movement of the nascent chain toward and away from the label at the translocon as two clearly defined steps. We then refine the values by global fitting of both translation and FRET time courses for different LepB constructs, using the kinetic model depicted in Fig 2C. For each chain length, we calculate its characteristic FRET value, which we denote as intrinsic fluorescence intensity (IFI) (Fig 2E). The validity of the IFI approach in identifying intermediates on a co-translational folding pathway was shown by force-profile analysis and molecular-dynamics simulations (Nissley & O'Brien, 2018; Kemp et al, 2019). The trend in FRET changes obtained from the IFI analysis is consistent with the endpoint FRET values (Fig 2E), but also takes into account the kinetics of translation and scans for the existence of other potential states that are not sampled by the end-level analysis with truncated mRNAs. The observed FRET changes provide the simplest description of the topogenesis of LepB TM1 as a head-on insertion of the nascent chain into the translocon and movement toward the periplasmic side to reach the N-out orientation.

## Co-translational insertion of a type II TM (EmrD)

As a model for a type II TM with the N-terminus of TM1 pointing into the cytosol, we use EmrD, a membrane protein containing 12 TM segments and three positive charges near the N-terminus of TM1. We analyze the co-translational insertion of nascent peptides comprising 50–135 N-terminal aa of EmrD (Fig 3), with the largest construct being sufficiently long for inversion and insertion of TM1 (Appendix Fig S4A). We also study a variant of EmrD, EmrD(–), in which three positively charged aa at the N-terminus of the nascent chain are replaced with negatively charged ones. The results obtained with the mutant are described in the following section, but are included in Fig 3 for easier comparison.

From exponential fitting of time courses (Fig 3A, Appendix Fig S5), the average translation rate is 2.5 aa/s (Appendix Table S2) independent of the presence of SecYEG (Appendix Fig S4B and C). We then build a kinetic model for EmrD synthesis which includes 134 elongation steps (Fig 3C). As with LepB, fits based on a linear model are unsatisfactory. Introducing a single pausing intermediate at codon 48, based on ribosome profiling data (Mohammad et al, 2016), provides satisfactory results (Appendix Fig S6). The elongation rate $k_{el}$ is 4.7 aa/s, close to that determined for LepB mRNA, whereas the kinetic parameters of the pausing intermediates are different (Fig 3C).

Next, we examine the co-translational EmrD insertion into the translocon by time-resolved FRET. Co-translational fluorescence changes of EmrD135 indicate approach of the N-terminus to the donor-in label position prior to donor-out (Appendix Fig S4D) as for LepB75, although no slow fluorescence decrease is apparent. We again measured fluorescence changes during insertion of different lengths of nascent peptide using a translocon with the donor label positioned on the cytoplasmic side (Fig 3E, Appendix Fig S4D–F). At a length of 50 aa, essentially no FRET change is observed (Fig 3E, black trace), similar to what is seen for the shortest LepB construct, LepB35 (Fig 2D). For nascent chains of 60–135 aa, a FRET increase is observed after about 10 s, with a moderate FRET increase for EmrD60 and higher FRET for EmrD70. EmrD85 shows a rapid FRET increase followed by a very slow decrease. The two longer EmrD constructs (110 and 135 aa), however, do not show a slow FRET decrease and reach about the same high-FRET level as reached with EmrD70. To determine which of these changes are co- and which post-translational, we fit the stopped-flow traces (Fig 3E) with delay-exponential functions. The resulting relaxation times (Appendix Table S2) indicate that the fluorescence changes observed with EmrD60 and EmrD70 coincide with translation and thus represent co-translational events. Similarly, the rapid upward phase observed with EmrD85 takes place during translation. In contrast, the observed very slow FRET decrease occurs after the synthesis of EmrD85 is completed. This indicates that the FRET decrease represents a post-translational event that is almost 10 times slower than translation and the initial FRET increase. Such a slow phase is not observed at other lengths of the nascent chain and probably represents an off-pathway state that forms when translation is stopped at this particular chain length. We then use the kinetic model for EmrD translation to fit the FRET changes accompanying the insertion of

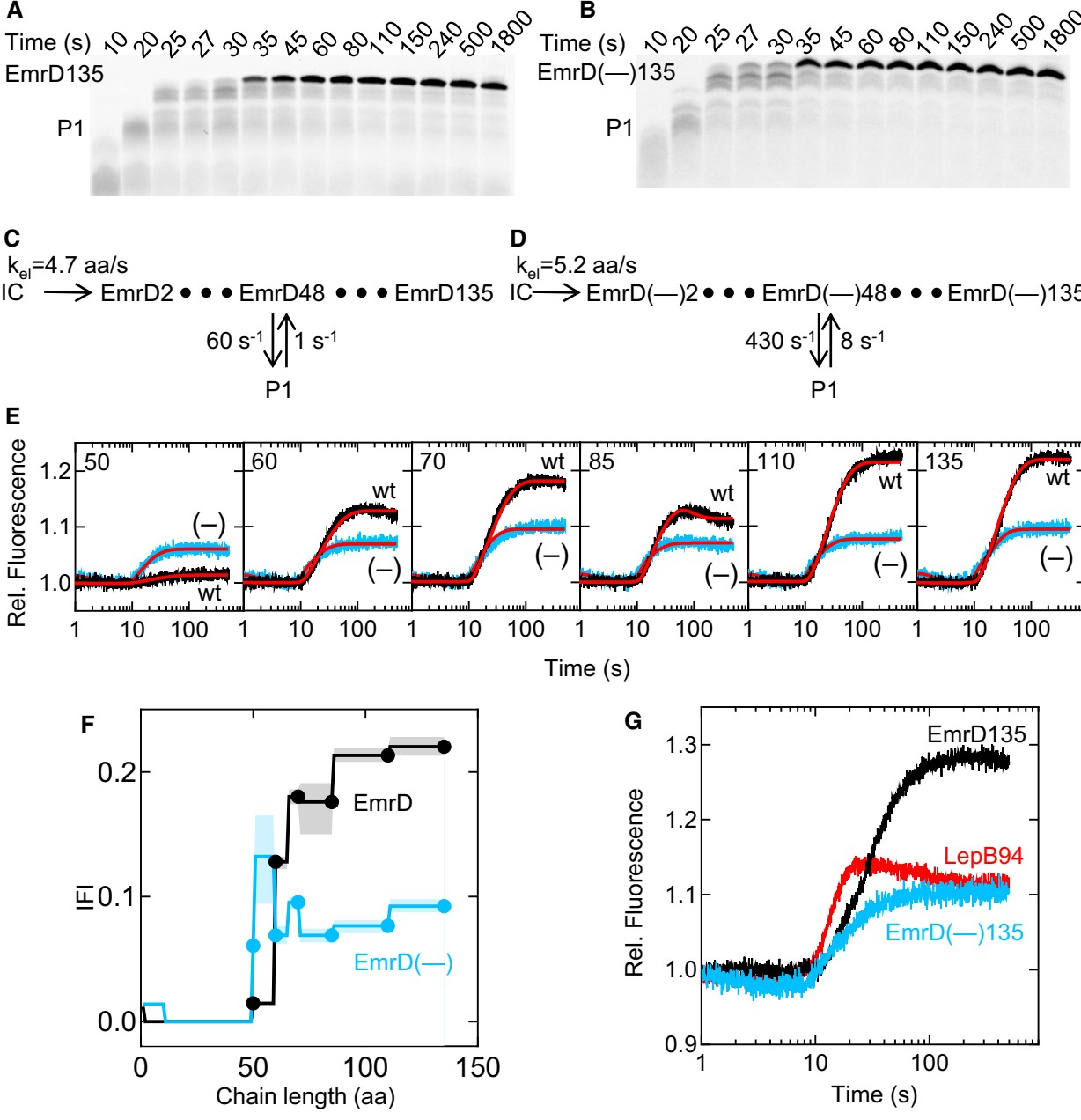

**Figure 3. Co-translational insertion of N-in membrane protein EmrD.**

A  Time course of EmrD135 translation. Translation was monitored by SDS–PAGE as in Fig 2A. For translation time courses of shorter mRNA constructs, see Appendix Fig S5.
B  Time course of EmrD(–)135 translation.
C  Kinetic model of EmrD synthesis. Ribosome stalling (peptide P1) is modeled at EmrD48.
D  Kinetic model of EmrD(–) synthesis. Stalling (peptide P1) is modeled at EmrD(–)48.
E  Stopped-flow time courses for translocon insertion of TM1 of EmrD (black) and EmrD(–) (blue) with mRNA constructs of varying chain length (50–135 aa).The FRET acceptor (Atto655) was placed at the N-terminus of the nascent peptide and the donor (Atto488) at position 111 (donor-in) of SecY. The results of global fitting are indicated (red lines).
F  Intrinsic fluorescence intensities (IFIs) calculated for EmrD (black) and EmrD(–) (blue). Error margins (shaded areas) are calculated using the FitSpace algorithm (Materials and Methods).
G  Stopped-flow traces for co-translational insertion of LepB94 (red, from Fig 2), EmrD135 (black, from Fig 4E), and EmrD(–)135 (blue, from Fig 4E), rescaled according to the translation efficiency.

EmrD. The co-translational IFI values reveal that the N-terminus of EmrD approaches the cytoplasmic side of the translocon rather late, when the nascent chain is between 60 and 70 aa in length (Fig 3F), compared to 50–52 aa of LepB (Fig 2E). The N-terminus of EmrD then remains near the cytoplasmic side of the translocon, consistent with the N-in topology of TM1.

## Charge reversal at the N-terminus switches TM topology

The N-in topology of membrane insertion generally is enforced by positively charged aa at the N-terminus of the nascent chain (von Heijne, 1989). EmrD has three positively charged aa near the N-terminus (Lys2, Arg3, and Lys5). In order to test when the positive-inside rule is established during co-translational insertion, we replaced the three aa with Glu, resulting in a negatively charged N-terminus of the mutant, EmrD(–), which is expected to reverse the $N_{out}/N_{in}$ ratio (Parks & Lamb, 1991). The kinetics of translation is not altered by the aa exchange (Fig 3A–D, Appendix Fig S5 and Appendix Table S3). However, the FRET changes are clearly different for wild-type EmrD and EmrD(–) (Fig 3E, blue traces). The delay-exponential fitting suggests a lack of post-translational rearrangement steps such as those observed with EmrD85; the FRET changes are co-translational for all chain lengths tested. Importantly, the FRET observed for EmrD(–)50 is much higher than that observed for wild-type EmrD50 (Fig 3E). This indicates that the positive charges at the N-terminus of EmrD are responsible for the delayed emergence of the nascent chain at the cytoplasmic side of the translocon. Surprisingly, the amplitude of the FRET change is lower for EmrD(–) than for EmrD, although the translation efficiency of the two mRNAs is identical. Moreover, the FRET at longer chain lengths of EmrD(–) is identical to that of LepB75, which adopts a stable N-out topology (Fig 3G).

There are two potential explanations for these findings. One possibility is that the N-terminus of the EmrD(–) constructs never approaches the cytoplasmic face of the translocon to the same extent as EmrD and LepB constructs do. Because the end-level FRET is similar for all EmrD(–) constructs, this model would imply that the nascent chain does not change its position relative to the label at the translocon as it grows from 50 to 135 aa in length. This is difficult to imagine, given the restricted space between the peptide exit and the translocon (Frauenfeld *et al*, 2011). Alternatively, the growing nascent chain of EmrD(–), which is expected to adopt an N-out topology due to the absence of positive charges in TM1, might follow a similar pathway as LepB, but because there is no pause in

the translation of EmrD(–) at the time when the high-FRET state is reached, the high-FRET intermediate does not accumulate. To account for the latter possibility, we performed fitting where IFI values for intermediates between aa 50 and 60 were allowed to increase beyond the end-level values (Fig 3F). This resulted in a fit that is statistically better than a more conservative fit where the high-FRET state was not permitted (Appendix Fig S7). The tendency in IFI values for EmrD(–) resembled that for LepB (Fig 2E), suggesting a similar topogenesis pathway. The difference between EmrD(–) and LepB is in the duration of the high-FRET state which corresponds to the incorporation of 9 aa and 24 aa, respectively, indicating that the orientation of EmrD(–) and LepB at the translocon, although grossly similar, may differ in detail.

## Protease accessibility of LepB and EmrD nascent chains

As an independent means to probe the nascent chain during translation, we monitored the accessibility of the nascent chain for proteinase K (PK) in a co-translational assay using radiolabeled N-terminal methionine (Materials and Methods). In the absence of translocon, the nascent chain of LepB is protected by the ribosome during the first 10 s of translation and on continued translation becomes exposed to PK (Fig 4A, Appendix Fig S8A and B). Fitting a delay-exponential function to the PK accessibility time course (Appendix Fig S8D) reveals a transit time of $16 \pm 2$ s for the emergence of the LepB nascent chain from the ribosome (Table 1). Emergence of the LepB nascent chain is, therefore, concomitant with the FRET increase observed during co-translational insertion of LepB75, which has a transit time of about 15 s.

For EmrD, the nascent chain becomes sensitive to PK digestion at the same time as LepB ($16 \pm 2$ s; Fig 4B and Table 1), but this occurs significantly earlier than the FRET increase observed for EmrD during co-translational insertion ($30 \pm 2$ s). This surprising result indicates that the nascent chain of EmrD is accessible to PK cleavage before the N-terminal FRET acceptor approaches the FRET donor at the cytoplasmic side of the translocon. This suggests that the N-terminus is directed away from the tunnel exit, and instead,

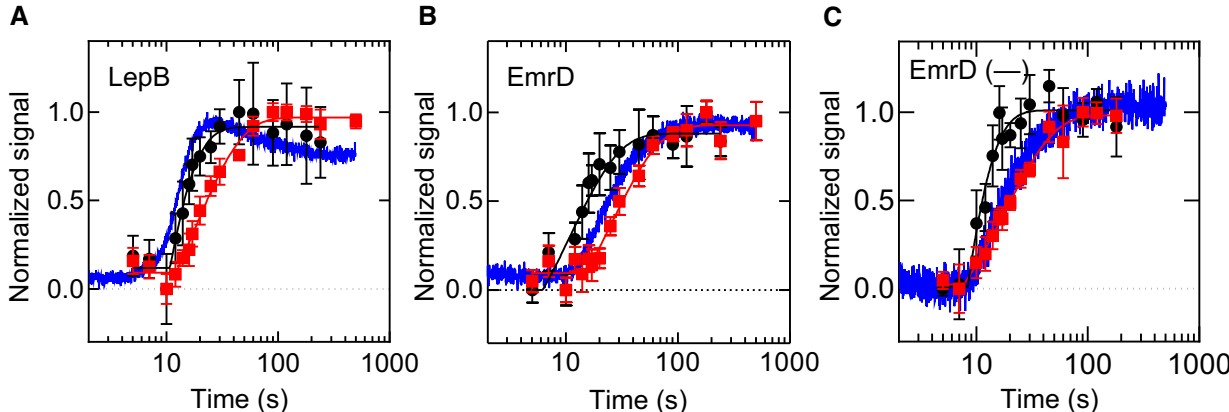

**Figure 4.** **Protection of nascent chains against PK digestion.**

A–C LepB (A), EmrD (B), and EmrD(–) (C) are translated for the times indicated and subjected to PK treatment (Materials and Methods). Sensitivity to PK in the absence of SecYEG (black circles) or in the presence of SecYEG (red squares), and co-translational translocon insertion monitored by FRET (blue trace) are depicted. Error bars represent standard deviations ($n \geq 3$), and the corresponding lines represent fits to delay-exponential functions.

**Table 1.  Transit times for translation, translocon insertion, and protection against PK digestion**

| Observable | τ (s) | | |
|---|---|---|---|
|  | **LepB** | **EmrD** | **EmrD(–)** |
| PK sensitivity | 16 ± 2 | 16 ± 2 | 13 ± 2 |
| FRET increase | 15 ± 1 | 30 ± 2 | 18 ± 3 |
| Protection by SecYEG | 30 ± 3 | 41 ± 4 | 26 ± 3 |
| FRET decrease | 50 ± 2 | – | – |

an internal position of the nascent chain is accessible to PK in the vestibule of the tunnel (see Discussion). The nascent chain of EmrD(–), on the other hand, becomes sensitive to PK at about the same time as the FRET increase is observed (Fig 4C). This indicates that positive charges at the N-terminus play a role in the delayed FRET increase relative to PK sensitivity for EmrD. The timing of translocon protection is similar for LepB and EmrD(–), but different for wild-type EmrD (Table 1). Considering the kinetic models for LepB and EmrD synthesis, the transit times indicate that LepB is stably inserted into the translocon when the nascent chain has reached a length of 68 aa (Materials and Methods), i.e., before TM2 is completely synthesized. By contrast, EmrD is protected at 80–120 aa, indicating that EmrD requires both TM1 and TM2 for insertion. In comparison, protection of EmrD(–) by SecYEG is similar to LepB and occurs at 56–84 aa, before TM2 has emerged from the ribosome. The transit times for EmrD(–) presented in Table 1 are similar to LepB rather than for wild-type EmrD and suggest a type I-like insertion of the variant.

## Discussion

The results of FRET and protease digestion experiments suggest a mechanism of co-translational topogenesis of type I and type II inner-membrane proteins (Fig 5). During synthesis of the type I protein LepB, TM1 moves head-on within the exit tunnel toward the exit port where it reaches the cytoplasmic side of the translocon. When the nascent chain is 35 aa in length, it is mostly protected by the ribosome and the acceptor-labeled N-terminus has not yet reached the donor-labeled translocon (no FRET). As the chain grows to 50 aa, which includes TM1 and the inter-TM linker, the N-terminus of the LepB nascent chain emerges from the exit tunnel and approaches the cytoplasmic side of the translocon, as indicated by the appearance of FRET (mid-FRET). At a length of about 50 aa, nascent chains are accessible to proteolysis, indicating that the conduit of the ribosome and translocon tunnel is not completely sealed and allows for the access of PK. The N-terminus of the growing peptide remains close to the donor label at the cytoplasmic face of the translocon as the nascent chain lengthens to 60 and 70 aa, and then moves toward the periplasmic side (Fig 5). The stable insertion of LepB TM1 into the translocon, which makes TM1 inaccessible for PK, occurs at about 68 aa. Remarkably, TM engagement with the translocon coincides with a translational pause at codon 68. A pause may provide a time window for the nascent peptide to equilibrate between the translocon pore and the membrane, thus helping to attain the correct TM1 topology. When the chain length

reaches 75 aa, the LepB N-terminus assumes a position at the periplasmic face of the translocon where it remains as the nascent chain grows further to 94 aa. At 75 aa, the nascent chain is distributed between the translocon (covering about 25 aa from the N-terminus) and the ribosome (50 aa), which should be easily accommodated, given the dimensions of the exit tunnel and the translocon pore. Thus, topogenesis of LepB TM1 can be described as a head-first movement through the exit tunnel–translocon conduit toward the periplasmic face of the translocon. It is less clear what happens with nascent chains longer than 94 aa, where the growing peptide accumulates, and how TM2 adopts the inverted N-in topology during ongoing translation. One possibility is that longer chains are extruded into the cytoplasm at the junction between the ribosome and the translocon and insert into the membrane at a later stage.

*In vivo* and *in silico* analyses of TM insertion using arrest-peptide-mediated stalling indicate that pulling forces act on the nascent chain at two distinct times during TM insertion. The first, smaller, pulling force is exerted when the TM is about 30 aa away from the PTC and the TM first reaches the interior of the translocon (Ismail *et al*, 2012; Niesen *et al*, 2018). This is in keeping with the FRET increase between LepB50 and LepB60 observed in the IFI analysis presented here. The second, stronger, pulling force is generated when the TM is about 40 amino acids from the PTC and is suggested to coincide with TM partitioning into the lipid bilayer (Ismail *et al*, 2012; Niesen *et al*, 2018), which would fit with the FRET decrease in the IFI values around LepB75 observed in this study.

In contrast to type I membrane proteins, which insert into the membrane in an N-out configuration, type II proteins insert into the membrane with their N-termini pointing into the cytoplasm. EmrD provides an example of how this can occur co-translationally. Although EmrD is synthesized at a similar rate as Lep B, its N-terminus does not come close to the reporter at the cytoplasmic face of the translocon until the nascent chain reaches a length of about 60 aa (Fig 5). At first glance, this appears to contradict the results of the PK digestion experiments, which indicate that EmrD and LepB nascent chains become sensitive to the protease at about the same time. We explain this observation by retention of the N-terminus within the exit tunnel such that it remains distant from the translocon (no FRET), while the growing peptide continues to move within the tunnel adopting a tail-first looped conformation. When the looped nascent peptide is extruded from the exit tunnel, internal parts of the nascent peptide, rather than the N-terminus, become susceptible to PK cleavage. The emerging tail-first looped conformation is compatible with the dimensions of the exit tunnel, which can occlude small protein domains of comparable size (Nilsson *et al*, 2015). Importantly, the picture changes when the positively charged residues at the N-terminus of EmrD are replaced with negative ones. The nascent peptide with negative charges at the N-terminus moves into the mid-FRET position earlier than native EmrD and at almost the same chain length as LepB. These results provide strong evidence suggesting that retention and inversion of EmrD TM1 are governed by electrostatic interactions. Positive charges at the N-terminus of nascent EmrD are important for retention and inversion to enter the membrane in an N-in orientation. It is often assumed that the positively charged N-terminus of the nascent peptide interacts with negatively charged phospholipids at the entrance to the translocon, leading to retention. However, the present data suggest that, at least for EmrD, the retention occurs at a relatively short

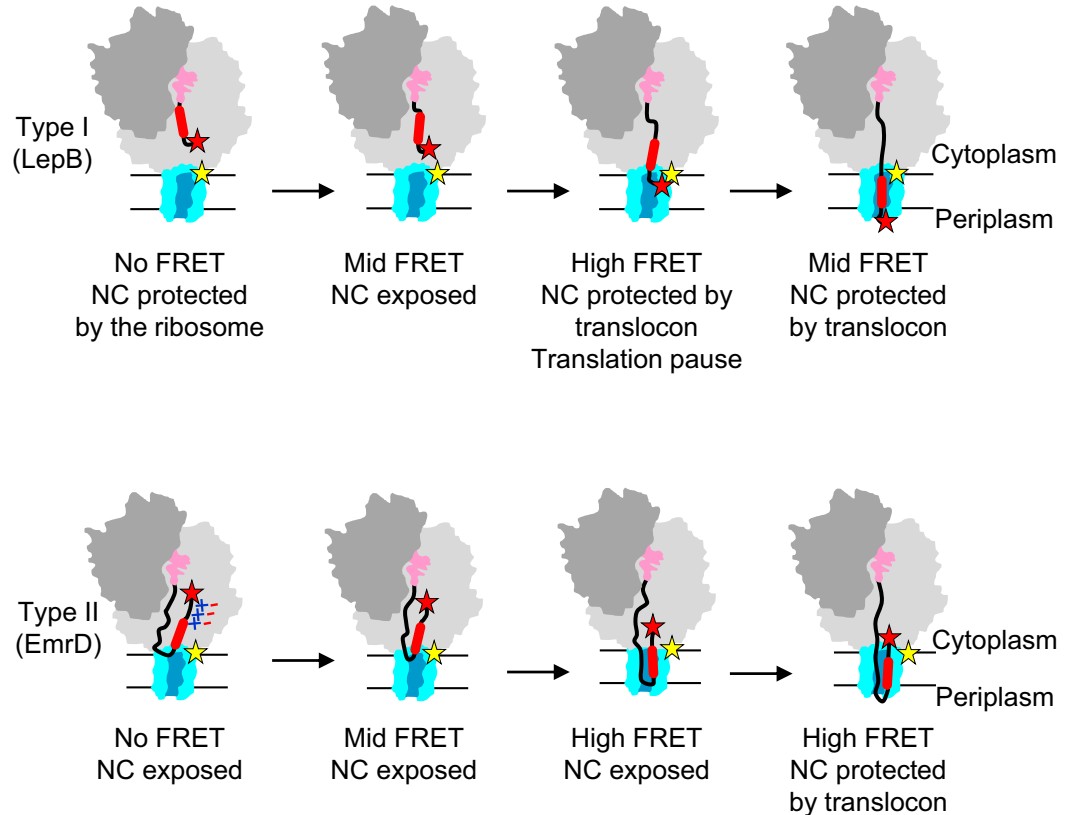

**Figure 5. Topogenesis of type I and type II membrane proteins.**

FRET labels are positioned at the N-terminus of the nascent chain (red star) and at the cytoplasmic face of the translocon (yellow star). NC, nascent chain. For further explanations, see text.

chain length when the N-terminus still resides in the exit tunnel and is located away from the translocon. Thus, a more likely explanation of our results is that positive charges at the N-terminus of the nascent peptide interact with negative charges of the rRNA of the ribosome tunnel wall, which causes retention of the N-terminus and inversion of TM1 within the tunnel upon continued peptide elongation. *In vivo,* the membrane potential may additionally influence topogenesis of membrane proteins in bacteria (Andersson & von Heijne, 1994; Cao *et al*, 1995; van der Laan *et al*, 2004; Knyazev *et al*, 2018). Movement of the nascent chain toward the tunnel exit and into the translocon requires that eventually the N-terminus is released from the electrostatic interactions in the tunnel. The release could result from a "push" exerted by the lengthening of the nascent chain by continued translation and/or the "pull" created by the insertion of the nascent TM segment into the translocon (Ismail *et al*, 2012).

Protease protection experiments suggest that EmrD remains accessible until it reaches a length of 80–120 aa, whereas LepB is protected by the translocon at a much shorter chain length (68 aa). This indicates that the interface between the ribosome exit port and the translocon is dynamic. At a chain length of about 100 aa, both TM1 and TM2 are about to emerge from the ribosome. It is thus possible that the labile ribosome–translocon interface reflects an accumulation of helical segments before they are stably inserted into the phospholipid phase. This protease-sensitive

state may result from phospholipid-surface-bound TMs which have been identified during spontaneous TM integration (Ulmschneider *et al*, 2014). The difference between a one-helix (LepB) and a two-helix (EmrD) insertion presents a mechanistic difference in membrane insertion of type I and type II membrane proteins. A labile interface may also allow parts of the nascent protein to accumulate at the cytoplasmic side of the translocon, allowing for the folding of non-membrane parts of inner-membrane proteins, such as their cytoplasmic domains.

The mechanism of type II TM topogenesis suggested by this study appears to contradict previous studies that identify an early N-out intermediate and suggest a late-inversion mechanism (Goder & Spiess, 2003; Devaraneni *et al*, 2011). Those studies were performed in eukaryotic systems, which may in part explain the difference. The lower part of the peptide exit tunnel is substantially narrower in eukaryotes than in bacteria (Dao Duc *et al*, 2019), which in eukaryotic ribosomes may preclude extensive reorientation of a TM within the tunnel and therefore externalize the inversion to a downstream component. In addition, differences may arise from the fact that those studies employ endpoint assays with truncated nascent chains that cannot distinguish co-translational pathway intermediates from those formed post-translationally or result from insertion of incomplete proteins. Since the present study focuses on the inaugural steps in membrane insertion, we note that later insertion and topological events may add significant complexity to the

mechanism. The topology of some membrane proteins, including dual-topology proteins, can be influenced by amino acid substitutions near or even at the C-terminus, indicating that topology is not necessarily fixed after the insertion of TM1 (Lu *et al*, 2000; Seppälä *et al*, 2010; Woodall *et al*, 2015; Fluman *et al*, 2017). The role of the ribosome and translation in these situations is not known. The orientation of model signal-anchor sequences, on the other hand, seems to rely on rapid synthesis and translocation of a C-terminal periplasmic domain (Goder & Spiess, 2003). For these model constructs, a kinetic competition between lipid integration and inversion of the signal-anchor sequence seems to be at play, with slower translation providing more time for inversion (Zhang & Miller, 2012; Niesen *et al*, 2017).

The present study clarifies the role of the ribosome and membrane protein synthesis in co-translational TM insertion and, in particular, TM topogenesis. We find that TM insertion is rapid and does not represent an independent kinetic step in membrane protein insertion when translation is ongoing. This is consistent with the finding that spontaneous membrane insertion of TMs can take place rapidly without translocon (Ulmschneider *et al*, 2014) or aided by YidC (200 s$^{-1}$; Winterfeld *et al*, 2013). The role of the ribosome during membrane protein insertion is to ensure that TM elements reach the translocon at the correct time and in the correct orientation. We see with EmrD85, for example, that stopping translation can yield an intermediate that undergoes a slow rearrangement into a state that resembles an N-out configuration. Participation of the ribosome in determining TM topology helps to explain how the positive-inside rule may be maintained during Sec-independent co-translational insertion in mitochondria (Oxa1 insertase), bacteria (YidC-only insertion), and in the ER (ER–membrane complex; Chitwood *et al*, 2018). Importantly, the timing of TM arrival at the translocon seems to be effectively encoded within the mRNA, since regulatory factors are not included in our *in vitro* system. Translocon insertion of LepB TM1 coincides with the translational pause in this study, suggesting that local translation may be fine-tuned for optimal insertion. Understanding the link between local translation rates and TM insertion/inversion will be central in the future understanding of co-translational membrane protein insertion and folding.

# Materials and Methods

## Materials

All experiments were performed at 37°C in buffer A: 20 mM Tris (pH 7.5), 70 mM $NH_4Cl$, 30 mM KCl, 3.5 mM $MgCl_2$, 8 mM putrescine, and 0.5 mM spermidine. Translation components from *Escherichia coli*, including ribosomes, initiation factors (IF1, IF2, and IF3), and elongation factors (EF-Tu, EF-Ts, and EF-G), were purified following published protocols (Wieden *et al*, 2002; Milon *et al*, 2007; Cunha *et al*, 2013). Total tRNA from *E. coli* (Roche) was aminoacylated as described elsewhere (Holtkamp *et al*, 2015) with inclusion of [$^{14}$C] Leu. Fluorescence-labeled initiator tRNA was prepared by reacting purified Met-tRNA$^{fMet}$ with excess Atto655-NHS ester (Atto-Tec) followed by purification via reverse-phase HPLC on a C-18 column (Merck) as described (Mittelstaet *et al*, 2013). Recombinant SecYEG variants and MSP1D1 were expressed and purified following established protocols (Ge *et al*, 2014; Draycheva *et al*, 2016). The SecY

(Q212C)EG and SecY(S111C)EG variants were prepared by site-directed mutagenesis of cysteine-less SecYEG, as described previously (Ge *et al*, 2014; Draycheva *et al*, 2016). Fluorescence labeling of the Cys variants of SecYEG was carried out with excess Atto488-maleimide (Atto-Tec); excess dye was removed by gel filtration (Sephadex G-25, GE Healthcare). Nanodiscs containing SecYEG were prepared from purified SecYEG (with or without a fluorescence label), MSP1D1 protein, and total *E. coli* lipids (Avanti Polar Lipids) and isolated after size-exclusion chromatography on a Superdex 200 column according to published protocols (Ge *et al*, 2014).

Genes encoding LepB or EmrD were amplified by PCR from *E. coli* (DH5α) genomic DNA and cloned into a pET24a vector by ligation-independent cloning. The EmrD(–) sequence was introduced by site-directed mutagenesis of the plasmid containing wild-type EmrD (EmrD(–): 5′ atg Gaa GAg caa GAa aac gtc aat... 3′). All DNA constructs were confirmed by sequencing (SeqLab). For each mRNA in this study, the corresponding DNA fragment was amplified by PCR starting from the T7 promoter and terminating with the last codon of the desired mRNA (no stop codon was included). The PCR product was then used as template for *in vitro* transcription with T7 RNA-polymerase as described (Holtkamp *et al*, 2015), and the mRNA was purified by anion exchange chromatography on a 5 ml HiTrap Q column (GE Healthcare). All experiments contained GTP (Jena Biosciences), phosphoenolpyruvate, and pyruvate kinase from rabbit muscle (Roche). Proteinase K from *Tritirachium album* was purchased from Sigma.

## Preparation of initiation complexes

Initiation complexes were prepared with 70S ribosomes, 1.5-fold excess of each initiation factor (IF1, IF2, and IF3), 10-fold excess mRNA, 2.5-fold excess initiator tRNA (Atto655-[$^3$H]Met-tRNA$^{fMet}$ or f[$^3$H]Met-tRNA$^{fMet}$), and 1 mM GTP. Initiation was carried out at 37°C for 1 h in buffer B: 20 mM Tris (pH 7.5), 70 mM $NH_4Cl$, 30 mM KCl, and 7 mM $MgCl_2$. All concentrations indicated are final concentrations. Initiation complexes were purified by centrifugation through a sucrose cushion (80 μl buffer B containing 1.1 M sucrose) in a TLA-100 rotor at 180,000 *g* for 1 h, dissolved in buffer A, flash-frozen, and stored at −80°C until use. The concentration of initiation complex was determined by scintillation counting of $^3$H-labeled methionine.

## *In vitro* translation

Translation was carried out at 37°C in buffer A with 25 nM purified initiation complex, 15 μM EF-Tu, 2 μM EF-G, 0.1 μM EF-Ts, total aminoacyl-tRNA containing 2.7 μM [$^{14}$C]Leu-tRNA$^{Leu}$ (estimated 15 μM total aa-tRNA), 1 mM GTP, 3 mM phosphoenolpyruvate, and 10 μg/ml pyruvate kinase, with or without 0.5 μM nanodisc-embedded SecYEG. To obtain translation time courses, translation was quenched by the addition of 2% ammonia (final concentration), and peptidyl tRNA was hydrolyzed for 30 min at 37°C. Samples were then dried in a SpeedVac and dissolved in SDS-loading buffer. Translation products were separated on 16% Tris-Tricine SDS–PAGE (Schagger, 2006) and imaged on an FLA-9000 fluorescence scanner (Fujifilm). Gels were analyzed using ImageJ software.

To determine translational efficiency, translation with Atto655-labeled initiation complexes was carried out in the presence of

0.5 μM SecYEG as above. The samples were overlaid on 1.1 M sucrose solution, and RNCs were then purified from free aminoacyl-tRNA by centrifugation at 180,000 $g$ in a TLA-100 rotor (Beckman) at 4°C for one hour. Pelleted RNCs were dissolved, the concentration of ribosomes was determined by UV absorbance ($\varepsilon_{260} = 43.5$ μM$^{-1}$cm$^{-1}$), and the nascent chain concentration was determined by counting $^3$H-Met and $^{14}$C-Leu. Translation efficiency, calculated as the ratio of nascent chain per ribosome, was determined to be 79%, 76%, and 96% for EmrD135, EmrD(–)135, and LepB94, respectively.

## Monitoring translation by FRET

Co-translational protein insertion into the translocon was carried out in a stopped-flow apparatus (SX-20MV, Applied Photophysics) by rapid mixing of initiation complex containing Atto655-labeled Met-tRNA$^{fMet}$ with a solution containing translation components and nanodisc-embedded Atto488-labeled SecYEG in the final concentrations indicated above. The donor fluorophore was excited by a blue LED (SX LED 470, Applied Photophysics), and emissions were filtered by RG665 long-pass and 535/50 BrightLine HC band-pass filters for the acceptor and donor channels, respectively. Fluorescence emissions in the acceptor and donor channels were measured by R2228 (red-sensitive) and R6095 photomultiplier tubes (Applied Photophysics), respectively. Five or six replicates of each experiment were performed and averaged prior to analysis.

## Protease accessibility

Proteinase K (PK) sensitivity of an N-terminal radiolabel (f[$^3$H] Met) on LepB, EmrD, and EmrD(–) was tested during translation of the respective 135 codon-long mRNAs using the same concentrations of factors as listed above. At indicated time points, PK was added (1.4 mg/ml final concentration) along with CaCl$_2$ (5 mM final concentration). After 10 s, the digestion was quenched by the addition of 0.1 reaction volumes of 5 M KOH. Samples were incubated for 30 min at 37°C, and peptides were precipitated by the addition of 3.5 volumes of 10% TCA and 30 min incubation on ice. Precipitated peptides were collected on 0.45-μm nitrocellulose filters (Sartorius) via suction filtration; filters were washed first with 10 ml of cold 5% TCA and then with 5 ml cold 30% isopropanol before scintillation counting. Emergence of the nascent chain from the exit tunnel of the ribosome was determined by subtracting the time course without PK from the time course without SecYEG, to correct for length-dependent precipitation efficiency of peptides:

$$\text{Emergence(t)} = \text{Counts}_{\text{minusPK}}(t) - \text{Counts}_{\text{minusSecYEG}}(t) \qquad (1)$$

Similarly, protection by SecYEG was determined by subtracting the time course without SecYEG from the time course with SecYEG:

$$\text{Protection(t)} = \text{Counts}_{\text{plusSecYEG}}(t) - \text{Counts}_{\text{minusSecYEG}}(t) \qquad (2)$$

## Data analysis

The following delay-exponential functions were fitted to time courses using TableCurve software:

$$y(t) = \begin{cases} y_0 & \text{if } t < \text{delay} \\ y_0 + (1 - \text{Amp})e^{-k_{app}(t-\text{delay})}, & \text{if } t > \text{delay}. \end{cases} \qquad (3)$$

$$y(t) = \begin{cases} y_0 & \text{if } t < \text{delay} \\ y_0 + (1 - \text{Amp}_1)e^{-k_{app,1}(t-\text{delay})} \\ \quad + (1 - \text{Amp}_2)e^{-k_{app,2}(t-\text{delay})}, & \text{if } t > \text{delay}. \end{cases} \qquad (4)$$

Transit times were computed by summing the inverse apparent rate constant with the delay time as follows:

$$\tau = \text{delay} + \frac{1}{k_{app}} \qquad (5)$$

Kinetic modeling and calculation of intrinsic fluorescence intensities (IFIs) were performed in KinTek Global Kinetic Explorer (Version 8; Johnson, 2009), as previously described (Mercier & Rodnina, 2018). In short, kinetic models were constructed describing nascent chain elongation at single amino acid resolution (Figs 2C and 3C and D), and pauses were modeled by including reversible equilibria. The rate of amino acid incorporation was assumed to be the same at each step in the mechanism. Alternatively, we used a model where each amino acid addition depends on the concentration of the respective aminoacyl-tRNA cognate to the given codon, as well as the concentrations of the near-cognate and non-cognate aminoacyl-tRNAs; the kinetic constants were taken from previous work (Rudorf *et al*, 2014). With either model, no satisfactory fits were obtained without introducing pauses. Including pausing allowed to fit the translation courses with either model; in the following, we used the simpler model with the uniform $k_{el}$.

Initially, each kinetic model was used in global fitting of translation time courses for six different lengths of mRNA in addition to one or two translation intermediates (pauses). Next, stopped-flow data were added to the fit, where the time-dependent fluorescence in each experiment was modeled by a linear combination of all nascent chain lengths weighted by their respective IFIs:

$$F(t) = \sum_{i=1}^{n} \text{IFI}_i \cdot [\text{RNC}_i](t) \qquad (6)$$

where $i$ represents the length of each nascent chain (in amino acids), $n$ is the number of codons in the mRNA, and [RNC$_i$](t) is the time-dependent concentration of RNC with nascent chain length $i$. As described previously (Mercier & Rodnina, 2018), IFIs were combined such that one IFI described a group of RNCs with different nascent chain lengths. In this work, we sought to minimize the number of IFIs in the fit and, in doing so, allow more robust error analysis on the model. Error analysis was performed by examining the dependence of the sum square residuals on each individual parameter using the 1D FitSpace algorithm in KinTek Explorer software (Johnson *et al*, 2009). In order to provide more conservative estimates of the errors, the chi-square threshold recommended by the software for computing upper and lower boundaries for each parameter was decreased to 0.98, which reflects a tenfold increase in the confidence interval. The final models each included eight IFIs, and inclusion of additional IFIs (intermediates) did not affect the overall IFI profiles, but rather altered the IFI values slightly and superseded computation of the confidence intervals.

## Calculation of nascent chain length

Global fitting of translation time courses and fluorescence stopped-flow experiments provided fitted values for the kinetic parameters describing translation of each mRNA. These kinetic parameters ($k_{el}$, $k_{pause}$, $k_{unpause}$) were used to compute the average transit time for translation of each codon as follows:

$$\tau_i = \begin{cases} \frac{1}{k_{el}} & \text{if no pause at codon i} \\ \frac{1-P\,(pause)}{k_{el}} + \frac{(k_{pause}+k_{el})\cdot P\,(pause)}{k_{unpause}\cdot k_{el}} & \text{if pause at codon i.} \end{cases} \tag{7}$$

where $\tau_i$ is the time required to translate codon i, $k_{el}$ is the nominal rate of translation (in aa/s), $k_{pause}$ is the forward rate of pausing at codon i, $k_{unpause}$ is the rate at which the paused conformation returns to the translation pathway, and P(pause) is the probability of pausing given by:

$$P(pause) = \frac{k_{pause}}{k_{pause} + k_{el}} \tag{8}$$

The total time required to translate from the start codon to codon n is therefore given by:

$$\tau_{translation,n} = \sum_{i=2}^{n} \tau_i \tag{9}$$

**Expanded View** for this article is available online.

## Acknowledgements

We thank Anna Pfeifer, Olaf Geintzer, Sandra Kappler, Christina Kothe, Theresia Niese, Tanja Wiles, Vanessa Herold, Franziska Hummel, Tessa Hübner, Puyan Nabizadeh-Ardekani, and Michael Zimmermann for expert technical assistance. The work was supported by a grant of the Deutsche Forschungsgemeinschaft (SFB1190 to M.V.R.)

## Author contributions

EM prepared the materials, performed the experiments, and analyzed the data. EM, WW, and MVR conceived the research and wrote the paper.

## Conflict of interest

The authors declare that they have no conflict of interest.

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
