## [Review Process File · The EMBO Journal]

Co-translational insertion and topogenesis of bacterial membrane proteins monitored in real time

Evan Mercier, Wolfgang Wintermeyer and Marina V. Rodnina.

Review timeline:

Submission date:	19 th November 2019
Editorial Decision:	16 th December 2019
Revision received:	7 th February 2020
Editorial Decision:	21 st February 2020
Revision received:	27 th February 2020
Accepted:	3 rd March 2020

Editor: Elisabetta Argenzio

Transaction Report:

1st Editorial Decision

16th December 2019

Thank you for submitting your manuscript entitled "Co-Translational Translocon Insertion and Topogenesis of Bacterial Membrane Proteins" [EMBOJ-2019-104054] to The EMBO Journal. Your study has been sent to three referees for evaluation, whose reviews are enclosed below.

As you can see, the referees find your study interesting and raise only a few points that should be addressed before they can support the publication of your work in The EMBO Journal. In addition, the referees give you suggestions on how to improve your manuscript.

Given the overall interest of your study, I would like to invite you to revise the manuscript in response to the referee reports. I should note that conclusively addressing these and all the other referees' points is essential for publication in The EMBO Journal.

REFEREE REPORTS

Referee #1:

In this paper, the initial processes of Type I- and Type II-transmembrane insertion were analyzed by FRET. The authors employed a reconstituted in vitro bacterial co-translational system using purified proteins and nanodisc technology, which provide a native-like membrane environment. The results of stopped-flow FRET revealed that the initial states of Type I- and Type II-transmembrane insertions are different. It was further demonstrated that by changing the positive charges in a Type II protein, its mechanism of insertion can be altered to that of Type I. A series of FRET data obtained in the paper by using various lengths of the nascent chain are clear and reliable. Additionally, the results of the protease protection assay supported the FRET data, and provides information about the exposure of the nascent chain. The observation of real-time membrane protein insertion reported in this study would provide a technique that would be useful for further

biophysical approaches. Therefore, I believe that the paper should be published as early as possible. However, one weak point of this paper is that it is not possible to clarify the complex processes of dynamic membrane insertion in detail only from the data obtained by FRET and the protease assay.

-Major concern

The title appears to suggest a review, which is inconsistent with the contents of the manuscript. I would suggest rephrasing the title, and it would be better if the title would be such that it explains the study concretely. For instance, the authors might consider rephrasing the title as 'Initiation of bacterial co-translational membrane protein insertion monitored by FRET'.

-Minor concerns

1. If the authors have already obtained the FRET data of EmrE using the donor attached to the periplasmic side of SecY, the data should be discussed as shown in Fig. 1C.
2. In the figure legends of Figures 2 and 3, please provide the information of the position of the donor to ensure easy understanding of the results.
3. Because this study used the nanodiscs, the nanodiscs should be illustrated accordingly, and should not be represented as plain membranes in Figures 1 and 4 to avoid misunderstanding.
4. Please mention that the membrane potential is related to the positive inside rule in the discussion.
5. The speed of membrane protein insertion has been estimated in the following paper by YidC. Perhaps it would be a good idea to refer to and cite it in the discussion.

Winterfeld et al., (2013) <https://doi.org/10.1371/journal.pone.0059023>

Referee #2:

This is a well written paper providing significant insight in the role of translation in the membrane topology acquisition of type I and II membrane proteins. The experiments are well executed and the conclusions reasonable.

I have a few suggestions that could be considered prior to final formulation of the manuscript.

1. I find the last sentence of the abstract very strong. The authors do not provide strong mechanistic understanding for this. Even the idea of the positive charges contributing to inversion of the chain's topology even prior to ribosome release is tentative at the moment.
2. Although the authors exhaustively exploit the single FRET pair used here, a second probe positioned at the periplasmic side of the translocator would have addressed some issues much more directly (e.g. by looking to gain of FRET because of the proximity to the periplasmic side probe).
3. The legends are extremely spartan. I think that both experimental conditions should be more clearly explained here.
4. How is the EmrD(-) gain of FRET structurally rationalised? What is it supposed to represent? Also replacing three positive charges with three negative ones is an obtrusive substitution including causing non-physiological perhaps repulsive events. Why haven't the authors made a simple alanine substitution?
5. p12, why is there no pause for EmrD(-) translation?
6. the protease experiments were done with a different set of membrane protein substrates. It would have been to cross-check directly the protease accessibility of the fluorescently-labelled substrates.
7. the decrease in FRET signal for the long LepB constructs is assumed to come from distancing from the cytoplasmic probe although it is not obvious why the signal drops gradually rather than staying on average the same. It would be a simple straight-forward experiment to just use a more elongated construct that could provide an inter probe distance of 10nm or so and lose the fluorescence completely.
8. with respect to the idea that positive charges at the N-termini would allow hairpins to form inside the ribosome:
 - a. does introduction of multiple positive charges in one of the LepB constructs analysed cause reversion of its topology?
 - b. it would have been a more clean comparison to use an N-in single TM protein rather than a type II with 2 TMs because that introduces additional topological complexity.
 - c. do all type II proteins have multiple positive charges at their N-termini? Is this a universal

feature? The positive inside rule also applies to internal segments.

d. for this mechanism to work there would be obvious negatively charged patches in the exit tunnel. Can this be discussed?

e. in the case of secretory proteins that also have (optional) positive charges at their N-termini and that apparently also need to form hairpins to expose the signal peptidase cleavage sites, this mechanism would not be applicable since many of them are secreted post-translationally. I guess this would imply that if the proposed ribosome mechanism is at play it would be one of different solutions.

9. p17, model stop-transfer sequences. Not clear to the general public what is meant.

Referee #3:

Mercier et al. have followed the interaction of the first transmembrane segment (TM) in the E. coli inner membrane proteins LepB and EmrD with the SecYEG translocon in real time using a previously described FRET-based assay and proteolysis experiments. Briefly, FRET reporters are placed at the N-terminus of the NC and at one of two locations in SecY, one near the cytoplasmic entry into the translocon channel and one near the periplasmic exit from the channel. Purified SecYEG translocon in nanodiscs are then bound to mRAN-programmed ribosomes, and a synchronized translation reaction is initiated by addition of elongation factors and charged tRNAs. Finally, the FRET signal is followed in real time.

For the type I protein LepB - which has the N terminus of TM1 located in the periplasm (N-out topology) - the FRET data indicate that the N-terminal end of TM1 reaches the cytoplasmic channel entry at a NC length of ~50 residues, and then moves progressively towards the periplasmic channel exit, which it reaches at NC length of ~75 residues. This is consistent with a "head-first" insertion pathway for LepB TM1.

For the type I protein EmrD - which has an N-in topology - the FRET data indicate that the N-terminal end of TM1 reaches the cytoplasmic channel entry at a NC length of ~60 residues and then remains in this location during further chain elongation, consistent with a "loop" insertion mechanism where the positively charged N-terminus is retained on the cytoplasmic side of the membrane at all times.

Finally, the mutant EmrD(-) in which the three positively charged residues on its N terminus have been replaced by negatively charged residues has been analyzed - presumably, TM1 inserts in the N-out orientation in this mutant. Here, the N-terminal end reaches the vicinity of the cytoplasmic channel entry somewhat earlier than in WT, but the FRET curve does not show the characteristic "increase-decrease" pattern that is diagnostic of head-first insertion, which makes the EmrD(-) results more difficult to interpret.

Comments

To my knowledge, this is the first report where the cotranslational insertion of an N-terminal TM into the SecYEG translocon has been followed in real time. The experiments are straight-forward, and uses a well-established in vitro translation approach and a previously developed kinetic modelling scheme. The study provides some important figures for when during translation the NC makes contact with the translocon, and offers strong support for the "head-first" and "loop" insertion mechanisms for N-out and N-in proteins, respectively. It also demonstrates that the time-scale of the insertion process is set by the translation rate, not by the insertion process itself. The paper definitely warrants publication in EMBO J.

I have a few concerns/corrections that I hope the authors can address in a revision:

- p.5, 2nd paragraph: the catalytic domain of LepB resides in the periplasm, not the cytoplasm.
- When comparing the LepB and EmrD results, it must be taken into consideration that the TM segment itself starts within one or two residues of the N terminus in LepB, but is ~10 residues distant from the N terminus of EmrD. To the extent that entry of the NC into the translocon channel depends on hydrophobic interactions between the TM and the translocon/membrane, EmrD TM1

would be expected to engage with the translocon at ~10 residues longer NC length than LepB TM1, regardless of whether the positively charged N terminus interacts with the ribosome or not.

- The EmrD(-) data show that the N-terminal charge has some influence on how the NC behaves as it exits the ribosome, but are not so easy to interpret since we don't know if TM1 in this case translocates across the membrane (as in LepB) or remains on the cytoplasmic side (as in EmrD). It would be good to know if the N-terminus of EmrD(-) approaches the periplasmic FRET reporter in SecYEG at longer NC lengths or not - if it has N-out topology, it should reach full FRET signal with the periplasmic reporter (which would bolster the authors suggestion that there is a "hidden" high-FRET state that is not visible in the FRET traces in Fig. 3E). Likewise, for completeness, it would be good if the authors could add the the protease protection profile for EmrD(-) to Fig. 4.

1st Revision - authors' response

7th February 2020

Referee #1:

In this paper, the initial processes of Type I- and Type II-transmembrane insertion were analyzed by FRET. The authors employed a reconstituted in vitro bacterial co-translational system using purified proteins and nanodisc technology, which provide a native-like membrane environment. The results of stopped-flow FRET revealed that the initial states of Type I- and Type II-transmembrane insertions are different. It was further demonstrated that by changing the positive charges in a Type II protein, its mechanism of insertion can be altered to that of Type I. A series of FRET data obtained in the paper by using various lengths of the nascent chain are clear and reliable. Additionally, the results of the protease protection assay supported the FRET data, and provides information about the exposure of the nascent chain. The observation of real-time membrane protein insertion reported in this study would provide a technique that would be useful for further biophysical approaches. Therefore, I believe that the paper should be published as early as possible.

However, one weak point of this paper is that it is not possible to clarify the complex processes of dynamic membrane insertion in detail only from the data obtained by FRET and the protease assay.

Reply: We agree with the reviewer that our methods do not clarify all details of membrane insertion dynamics. However, combining two independent methods, i.e., co-translational FRET and protease protection, clarifies the extent to which the ensemble behavior changes as the nascent chain elongates and support the mechanism presented. A more detailed investigation of nascent-chain dynamics will require single-molecule techniques.

-Major concern

The title appears to suggest a review, which is inconsistent with the contents of the manuscript. I would suggest rephrasing the title, and it would be better if the title would be such that it explains the study concretely. For instance, the authors might consider rephrasing the title as 'Initiation of bacterial co-translational membrane protein insertion monitored by FRET'.

Reply: As suggested by the referee, we have changed the title to indicate that this is an experimental paper and not a review:

Co-translational insertion and topogenesis of bacterial membrane proteins monitored in real time

-Minor concerns

1. If the authors have already obtained the FRET data of EmrE using the donor attached to the periplasmic side of SecY, the data should be discussed as shown in Fig. 1C.

Reply: We have performed the suggested experiment. The data are now included in Appendix Figure S4 D and have been introduced into the results section on p. 11 which now reads “Co-translational fluorescence changes of EmrD135 indicate approach of the N-terminus to the donor-in label position prior to donor-out (Appendix Fig S4D) as for LepB75, although no slow fluorescence decrease is apparent.”

2. In the figure legends of Figures 2 and 3, please provide the information of the position of the donor to ensure easy understanding of the results.

Reply: The legends of Figs. 2 and 3 are expanded accordingly.

3. Because this study used the nanodiscs, the nanodiscs should be illustrated accordingly, and should not be represented as plain membranes in Figures 1 and 4 to avoid misunderstanding.

Reply: The structure of the nanodisc with inserted translocon, as determined by cryo-EM of an RNC-translocon complex is now depicted in Fig. 1B along with the positions of the donor label. To avoid overloading the figures, we have kept the schematic membrane in Fig. 1A and Fig. 5.

4. Please mention that the membrane potential is related to the positive inside rule in the discussion.

Reply: The role of the membrane potential is now mentioned in the Discussion on p. 17 along with pertinent references: “In vivo, the membrane potential may additionally influence topogenesis of membrane proteins in bacteria (Andersson & von Heijne, 1994, Cao et al., 1995, Knyazev et al., 2018, van der Laan et al., 2004)”.

5. The speed of membrane protein insertion has been estimated in the following paper by YidC. Perhaps it would be a good idea to refer to and cite it in the discussion.

Winterfeld et al., (2013) <https://doi.org/10.1371/journal.pone.0059023>

Reply: We have included this finding and the pertinent reference in the Discussion on p. 19.

Referee #2:

This is a well written paper providing significant insight in the role of translation in the membrane topology acquisition of type I and II membrane proteins. The experiments are well executed and the conclusions reasonable.

I have a few suggestions that could be considered prior to final formulation of the manuscript.

1. I find the last sentence of the abstract very strong. The authors do not provide strong mechanistic understanding for this. Even the idea of the positive charges contributing to inversion of the chain's topology even prior to ribosome release is tentative at the moment.

Reply: We have changed the last sentence of the abstract as suggested (p. 2).

2. Although the authors exhaustively exploit the single FRET pair used here, a second probe positioned at the periplasmic side of the translocator would have addressed some issues much more directly (e.g. by looking to gain of FRET because of the proximity to the periplasmic side probe).

Reply: We have performed co-translational insertion of Atto655-EmrD 135 with donor-out labeled SecYEG and present the results in Appendix Figure S4D and p. 11 ("Co-translational fluorescence changes of EmrD135 indicate approach of the N-terminus to the donor-in label position prior to donor-out (Appendix Fig S4D) as for LepB75, although no slow fluorescence decrease is apparent"). The donor-out traces with LepB 75 (Figure 1C) and EmrD 135 (Appendix Figure S4D) are less informative than donor-in, as they show only fluorescence increases (not an increase/decrease). This is not inconsistent with N-out insertion of LepB nor N-in insertion of EmrD, but it does not provide additional mechanistic insights. For this reason the in-depth analysis is focused on the donor-in data.

3. The legends are extremely spartan. I think that both experimental conditions should be more clearly explained here.

Reply: The legends have been expanded as suggested. Additions have been made in the legends of Figs. 1, 2 and 3. In particular the positions of fluorescence probes are now explicitly defined.

4. How is the EmrD(-) gain of FRET structurally rationalised? What is it supposed to represent?

Reply: Insertion of EmrD(-) is type I-like as it appears to insert without inversion. To make this clear, we have expanded the pertinent sentence on p. 13-14 as follows: "The tendency in IFI values for EmrD(-) resembled that for LepB (Fig 2E), suggesting a similar topogenesis pathway."

Also replacing three positive charges with three negative ones is an obtrusive substitution including causing non-physiological perhaps repulsive events. Why haven't the authors made a simple alanine substitution?

Reply: A net N-terminal charge of zero (which would be attainable by three alanine substitutions in EmrD wt), has been shown in other constructs to yield some N-in insertion, but the Nout/Nin ratio can be increased further by using a negatively-charged N-terminus (Parks and Lamb, Cell 1991) (sentence with reference added on p. 12). We have therefore adopted the charge reversal approach which was expected to provide a high probability of seeing an effect, and found that it worked. We also note on p. 13: “The kinetics of translation is not altered significantly by the aa exchange”

5. p12, why is there no pause for EmrD(-) translation?

Reply: There is a pause for EmrD(-) translation before codon 50 (at or near codon 48, Figure 3), the same as with EmrD. Because the pause does not occur at the high-FRET state (between codon 50 and 60), the high-FRET state is not obvious in the FRET-time course. This has been clarified on p 10 and 13: “...because there is no pause in the translation of EmrD(-) at the time when the high-FRET state is formed, the high-FRET intermediate does not accumulate”.

6. the protease experiments were done with a different set of membrane protein substrates. It would have been to cross-check directly the protease accessibility of the fluorescently-labelled substrates.

Reply: Unfortunately, this cannot be tested directly since filtration of fluorescence-labeled peptides is not possible due to the adsorption of fluorescence-labeled methionine to the filters. However, we have shown previously that there are no observable differences in translation kinetics when fMet-tRNA^{fMet} is replaced with fluorescence-labeled Met-tRNA^{fMet} (Holtkamp et al, Science 2015).

7. the decrease in FRET signal for the long LepB constructs is assumed to come from distancing from the cytoplasmic probe although it is not obvious why the signal drops gradually rather than staying on average the same.

Reply: The pause in translation that coincides with the formation of the high-FRET state results in a quasi-synchronization of the ribosomes at this point in translation. This is why we can observe both the high-FRET intermediate and the subsequent drop of the signal. We have added a pertinent sentence on p. 10: “We note that the pausing site at position 68 results in the accumulation of a high-FRET intermediate; this pause allows us to monitor the movement of the nascent chain towards and away from the label at the translocon as two clearly defined steps.” The gradual fluorescence decrease of LepB 75 coincides with the unpausing rate, indicating that the FRET decrease is co-translational, i.e. rate-limited by translation of LepB75. Without a pause at exactly this stage of translation, the signal would in fact stay on average the same, as observed for ErmD(-) (p. 13).

It would be a simple straight-forward experiment to just use a more elongated construct that could provide Ann inter probe distance of 10nm or so and lose the fluorescence completely.

Reply: Unfortunately this experiment is less straight-forward than one would expect. We have performed these experiments with a longer natural sequence of

LepB. We observed additional post-translational fluorescence changes which cannot be interpreted without understanding the insertion of TM2, which is beyond the scope of the current study. We have opted not to use longer constructs from which TM2 has been deleted as this would result in a totally unnatural sequence.

8. with respect to the idea that positive charges at the N-termini would allow hairpins to form inside the ribosome:
- does introduction of multiple positive charges in one of the LepB constructs analysed cause reversion of its topology?

Reply: These very experiments demonstrating inversion of LepB topology were presented in the seminal work by von Heijne regarding the positive inside rule (von Heijne Nature 1989) (cited on p. 12).

- it would have been a more clean comparison to use an N-in single TM protein rather than a type II with 2 Tis because that introduces additional topological complexity.

Reply: We have chosen LepB because it has been extensively studied by others and EmrD because it has features common to typical membrane protein. We wanted to use naturally occurring membrane protein sequences here, and found it difficult to find two sequences with single TMs at a similar locations but inserting in different topologies.

- do all type II proteins have multiple positive charges at their N-termini? Is this a universal feature? The positive inside rule also applies to internal segments.

Reply: Most, but not all type II proteins have multiple positive charges near the N-terminus and the positive inside rule also applies to internal segments. However, in the latter case, the neighboring helices also play a role in topogenesis.

- for this mechanism to work there would be obvious negatively charged patches in the exit tunnel. Can this be discussed?

Reply: The peptide exit tunnel is composed primarily of rRNA and therefore has negative patches along its entire length. This detail has been added to the Discussion on p. 17.

- in the case of secretory proteins that also have (optional) positive charges at their N-termini and that apparently also need to form hairpins to expose the signal peptidase cleavage sites, this mechanism would not be applicable since many of them are secreted post-translationally. I guess this would imply that if the proposed ribosome mechanism is at play it would be one of different solutions.

Reply: SRP-dependent secretory proteins are targeted to the membrane co-translationally, and their (cleavable) signal sequences presumably insert N-in by the mechanism we propose here. Proteins that are secreted post-translationally (for instance via SecA in bacteria) are not retained in the membrane.

9. p17, model stop-transfer sequences. Not clear to the general public what is meant.

Reply: We have replaced “stop-transfer sequence” with the more commonly used “signal-anchor sequence” (p. 19).

Referee #3:

Mercier et al. have followed the interaction of the first transmembrane segment (TM) in the *E. coli* inner membrane proteins LepB and EmrD with the SecYEG translocon in real time using a previously described FRET-based assay and proteolysis experiments. Briefly, FRET reporters are placed at the N-terminus of the NC and at one of two locations in SecY, one near the cytoplasmic entry into the translocon channel and one near the periplasmic exit from the channel. Purified SecYEG translocon in nanodiscs are then bound to mRAN-programmed ribosomes, and a synchronized translation reaction is initiated by addition of elongation factors and charged tRNAs. Finally, the FRET signal is followed in real time.

For the type I protein LepB - which has the N terminus of TM1 located in the periplasm (N-out topology) - the FRET data indicate that the N-terminal end of TM1 reaches the cytoplasmic channel entry at a NC length of ~50 residues, and then moves progressively towards the periplasmic channel exit, which it reaches at NC length of ~75 residues. This is consistent with a "head-first" insertion pathway for LepB TM1.

For the type I protein EmrD - which has an N-in topology - the FRET data indicate that the N-terminal end of TM1 reaches the cytoplasmic channel entry at a NC length of ~60 residues and then remains in this location during further chain elongation, consistent with a "loop" insertion mechanism where the positively charged N-terminus is retained on the cytoplasmic side of the membrane at all times.

Finally, the mutant EmrD(-) in which the three positively charged residues on its N terminus have been replaced by negatively charged residues has been analyzed - presumably, TM1 inserts in the N-out orientation in this mutant. Here, the N-terminal end reaches the vicinity of the cytoplasmic channel entry somewhat earlier than in WT, but the FRET curve does not show the characteristic "increase-decrease" pattern that is diagnostic of head-first insertion, which makes the EmrD(-) results more difficult to interpret.

Comments

To my knowledge, this is the first report where the cotranslational insertion of an N-terminal TM into the SecYEG translocon has been followed in real time. The experiments are straight-forward, and uses a well-established in vitro translation approach and a previously developed kinetic modelling scheme. The study provides some important figures for when during translation the NC makes contact with the translocon, and offers strong support for the "head-first" and "loop" insertion mechanisms for N-out and N-in proteins, respectively. It also demonstrates that the time-scale of the insertion process is set by the translation rate, not by the insertion process itself. The paper definitely warrants publication in EMBO J.

I have a few concerns/corrections that I hope the authors can address in a revision:

1. - p.5, 2nd paragraph: the catalytic domain of LepB resides in the periplasm, not the cytoplasm.

Reply: This has been corrected (p. 6).

2. - When comparing the LepB and EmrD results, it must be taken into consideration that the TM segment itself starts within one or two residues of the N terminus in LepB, but is ~10 residues distant from the N terminus of EmrD. To the extent that entry of the NC into the translocon channel depends on hydrophobic interactions between the TM and the translocon/membrane, EmrD TM1 would be expected to engage with the translocon at ~10 residues longer NC length than LepB TM1, regardless of whether the positively charged N terminus interacts with the ribosome or not.

Reply: TM1 of LepB starts at amino acid 4, and TM1 of EmrD starts at amino acid 10 as indicated in Appendix Figures S2 and S4. The FRET increase we observe likely starts shortly after the N-terminus passes the constriction site, which occurs prior to engagement of TM1 with the translocon (Appendix Figure S1). The difference we observe in the onset of FRET increase between LepB and EmrD (50 vs. 60 aa) reflects approach of the N-terminus to the cytoplasmic face of the translocon, rather than engagement. However, the maximum fluorescence change, which is probably closer to engagement, is observed at longer chain lengths (at least 60 aa for LepB and more than 85 for EmrD). This 25 aa difference is larger than the difference in the lengths of the N-termini preceding TM1 (6 aa). This is now discussed in more detail on p. 15: "...the transit times indicate that LepB is stably inserted into the translocon when the nascent chain has reached a length of 68 aa (Materials and Methods), i.e., before TM2 is completely synthesized. By contrast, EmrD is protected at 80-120 aa, indicating that EmrD requires both TM1 and TM2 for insertion. In comparison, protection of EmrD(-) by SecYEG is similar to LepB and occurs at 56-84 aa, before TM2 has emerged from the ribosome. The transit times for EmrD(-) presented in Table 1 are similar to LepB rather than EmrD wt, and suggest a type I-like insertion of the variant".

3. The EmrD(-) data show that the N-terminal charge has some influence on how the NC behaves as it exits the ribosome, but are not so easy to interpret since we don't know if TM1 in this case translocates across the membrane (as in LepB) or remains on the cytoplasmic side (as in EmrD). It would be good to know if the N-terminus of EmrD(-) approaches the periplasmic FRET reporter in SecYEG at longer NC lengths or not - if it has N-out topology, it should reach full FRET signal with the periplasmic reporter (which would bolster the authors suggestion that there is a "hidden" high-FRET state that is not visible in the FRET traces in Fig. 3E).

Reply: We have performed this experiment and we do observe a gain of FRET as the nascent chain approaches the donor-out label. This occurs later than the increase for the donor-in label (as expected). Unfortunately, due to the lack of additional fluorescence phases, this trace does not yield more kinetic information than the trace obtained with the donor-in label. As to the full FRET signal, precise quantification of FRET requires single-molecule FRET, rather than relative fluorescence, and is outside the scope of the current study. Type I insertion of the EmrD(-) variant is also supported by the protease protection experiment.

4. Likewise, for completeness, it would be good if the authors could add the the protease protection profile for EmrD(-) to Fig. 4.

Reply: As suggested, we have performed additional protease protection experiments with EmrD (-). The data have been added to Figure 4 and Table 1 and are described in the text (p. 14). The protease protection looks similar to LepB, which also inserts N-out. Page 14, bottom: “The nascent chain of EmrD(-), on the other hand, becomes sensitive to PK at about the same time as the FRET increase is observed. This indicates that positive charges at the N-terminus play a role in the delayed FRET increase relative to PK sensitivity for EmrD. The timing of translocon protection is similar for LepB and EmrD(-), but different for EmrD wt (Table 1)“. To better describe the data in Table 1, we have added one more sentence at the end of Results (p. 15, top): “In comparison, protection of EmrD(-) by SecYEG is similar to LepB and occurs at 56-84 aa, before TM2 has emerged from the ribosome. The transit times for EmrD(-) presented in Table 1 are similar to LepB rather than for wild-type EmrD, and suggest a type I-like insertion of the variant”.

2nd Editorial Decision

21st February 2020

Thank you for submitting a revised version of your manuscript. I have checked it carefully and found that all referees' points have been successfully addressed.

However, there are a few editorial issues concerning the text and the figures that I need you to address before we can officially accept the manuscript.

Marina Rodnina

EMBO J

Manuscript Number: EMBOJ-2019-104054